# Prevention of Prosthetic Joint Infection: From Traditional Approaches towards Quality Improvement and Data Mining

**DOI:** 10.3390/jcm9072190

**Published:** 2020-07-11

**Authors:** Jiri Gallo, Eva Nieslanikova

**Affiliations:** Department of Orthopaedics, Faculty of Medicine and Dentistry, Palacky University Olomouc, University Hospital Olomouc, I. P. Pavlova 6, 77900 Olomouc, Czech Republic; evanieslanikova@seznam.cz

**Keywords:** biomaterial-associated infection, prosthetic joint infection, preventative measures, at-risk patient, bacterial contamination, anti-adhesive, antibacterial surface treatment, quality improvement, machine learning

## Abstract

A projected increased use of total joint arthroplasties will naturally result in a related increase in the number of prosthetic joint infections (PJIs). Suppression of the local peri-implant immune response counters efforts to eradicate bacteria, allowing the formation of biofilms and compromising preventive measures taken in the operating room. For these reasons, the prevention of PJI should focus concurrently on the following targets: (i) identifying at-risk patients; (ii) reducing “bacterial load” perioperatively; (iii) creating an antibacterial/antibiofilm environment at the site of surgery; and (iv) stimulating the local immune response. Despite considerable recent progress made in experimental and clinical research, a large discrepancy persists between proposed and clinically implemented preventative strategies. The ultimate anti-infective strategy lies in an optimal combination of all preventative approaches into a single “clinical pack”, applied rigorously in all settings involving prosthetic joint implantation. In addition, “anti-infective” implants might be a choice in patients who have an increased risk for PJI. However, further progress in the prevention of PJI is not imaginable without a close commitment to using quality improvement tools in combination with continual data mining, reflecting the efficacy of the preventative strategy in a particular clinical setting.

## 1. Introduction

Prosthetic joint infection (PJI) is a disastrous complication of modern orthopedic surgery, frequently leading to prolonged morbidity and even to increased mortality [1,2,3]. Moreover, therapy for PJI is associated with enormous costs [4,5,6]. Although international efforts to minimize the risk of these infections are ongoing, PJIs continue to develop in alarming numbers. Current estimates of the rate of PJI range between 0.5% and 2.4% of primary hip and knee arthroplasties [7], and PJI can complicate up to 20% of revision arthroplasties [8,9]. Some authors have suggested that these numbers for primary total joint arthroplasty are both underestimated and increasing [10]. Gram-positive pathogens are thought to cause most PJIs, with staphylococci topping the list [11]. Indeed, the prevalence of methicillin-resistant *S. aureus* (MRSA) and *S. epidermidis* (MRSE) is increasing [12,13,14]. The caveat is that the microbiology associated with these events can be dynamic and difficult to capture. Results can depend on the location of the prosthetic joint (e.g., knee, hip, shoulder) and perhaps also on the type of PJI and related comorbidities [13,14,15]. Furthermore, the methods used to identify causative agents can vary, and the geographical location can be a source of microbial variability [16]. PJIs can be early (within 3 months), delayed (3–12 months), and late (>12 months). Early and delayed PJIs are associated with direct contamination at the time of surgery, whereas late/hematogenous PJIs are associated with blood delivery of infective agents.

The aim of this narrative review is to summarize the current knowledge in the field of PJI prevention. A particular emphasis is placed on methods that might integrate individual measurements.

### 1.1. Key Steps Towards PJI

PJI results from a complex interplay of numerous factors [17] that lead to an inability of periprosthetic immune cells to protect implant surfaces and periprosthetic tissues from bacterial colonization. For this reason, Gristina et al. proposed the concept of a “race for the surface” in which the host and bacterial cells compete in determining the ultimate fate of the implant [18]. This proposal has stimulated technological and biomaterial progress in ensuring clean operating rooms, appropriate skin preparation, antibiotic prophylaxis, and development of an “anti-infective” implant [19,20]. However, the idea is quite basic in comparison with current knowledge about the complexity of host–bacteria interactions [21]. Other factors that are, to some extent, out of our hands may play a role in the induction of dysbiosis or loss of a tolerance around an implant [13,22].

The most destabilizing factor is the ability of bacteria to adhere to and survive on virtually all natural and synthetic surfaces. Immediately after placement, a conditional protein film can develop, deriving from a reservoir of receptors for bacterial adhesive ligands that mediate the adhesion of free-floating bacteria to the surface of the biomaterial [23]. Several distinct classes of surface proteins have been identified as participating in firm adhesion of *S. aureus* microcolonies to a biomaterial and to each other [24]. The environmental and surface characteristics of a biomaterial, including surface roughness, hydrophobicity, and electrostatic charge, play conditional roles [25,26].

Once firmly attached to the surface of an implant, the microorganisms initiate *biofilm formation*, which is the creation of a “bacterial tissue” [23]. The formation of a biofilm is under the control of multiple genes, including those responsible for the secretion of protective slime and installation of quorum-sensing that enables communication inside the biofilm [23]. Mature biofilm of mono/polymicrobial origin makes bacteria extremely resistant to both the host immune system and antibiotic diffusion [27]. The transition between reversible and irreversible phases of biofilm formation, coupled with phenotypic change, is the last window of opportunity for clinically reasonable preventative measures (Figure 1).

In addition to biofilm-based PJI, the intracellular occurrence of invasive bacteria can also be part of the PJI presentation, contributing to the clinical variance observed in PJI. Intracellular bacteria including *Staphylococci* spp. can live in cytoplasm (cytoplasmic vacuoles) [28], and some pathogens can even invade the intranuclear environment [29]. Despite this intracellular umbrella, these bacteria cannot escape from the cells of innate immunity and can, for instance, be identified through a specific sensing mechanism of inflammasomes [30,31]. Moreover, bacteria have been detected living in the net of the bone canaliculi [32], and other places may serve as niches for bacteria inside the host. Details on host–intracellular pathogen interactions have been given elsewhere [33,34].

### 1.2. Why Has PJI Not Been Eradicated Despite Modern Preventative Measures?

The complete eradication of infections associated with indwelling implants is an unrealized dream of generations of orthopedic surgeons and scientists. The explanation for the inability to resolve this problem is complex. First, PJI may be a manifestation of host–bacteria interactions that differ from those traditionally assumed to result from intraoperative colonization of implant surfaces (conceptual limits). The current strategy is based on defending against bacterial invasion. However, bacteria may initially be nonpathogenic relative to the periprosthetic environment until some stimuli (miscommunication) shifts the host–bacteria interaction and triggers PJI development. Second, there may be a failure to eliminate a significant bacterial load during a surgery (organizational limits). Irregularities associated with routine processes in the operating room or perioperatively could underlie or facilitate PJI. We can never exclude the chance associated with day-to-day life (unintended systemic and individual mistakes). Third, some studies may overestimate the impact of a preventative intervention (scientific limits, biases) [35]. We have neglected the rule of replication in the pursuit of priority results, eliding the scientific reality that only a repeatedly verified result becomes a valuable result [36]. The key question along these lines is the extent to which we can rely on published data driving the preventative guidelines implemented in clinical practice. In most clinical studies, the implementation of an intervention is associated with any observed reduction in PJI. If this association were real in each of these studies, the combined absolute risk reduction for a set of particular interventions performed almost simultaneously (perioperatively) would be expected to shift PJI risk almost to zero. However, clinical data do not bear out this prediction [37]. The explanation for this may be that these studies are not appropriately designed to demonstrate the effect of these interventions, in part because of poor control of many non-inveterventional but clinically relevant factors.

## 2. Preoperative Strategies

The success of bacteria depends not only on the size of the bacterial load (all related to good surgical practice) but also on the ability of a host to eradicate them. For this reason, assessment of a patient’s individual risk for PJI is crucial even if the true potential for modifying some risk parameters in individual cases may be limited.

### 2.1. At-Risk Patients

A patient’s medical history can be a source of relevant information for calculating preoperative risk for the postoperative development of PJI. The presence of comorbidities and medications related to decreased effectiveness of the innate immune system is important for such analysis (Table 1). However, no sound mathematical link has been established with PJI development in most conditions. In addition, we do not know all the parameters for the optimization of a risk factor or the contribution of such optimization to the absolute risk reduction of PJI.

Some studies have shown that carriers of MRSA and MRSE could be at increased risk for PJI [56]. This finding has triggered a wide range of research activities examining the efficacy of preoperative decolonization [57]. Infections at other sites (organs) distant from the joint of interest also can contribute to the development of PJI and should be eradicated before total joint arthroplasty (TJA). In some institutions, clinicians target dental, urinary tract, gynecological, and soft-tissue infections, including of the skin.

### 2.2. Preventative Strategies for At-Risk Patients

The most effective preventative measure would be to restrict candidacy for TJA in patients who are at high risk for PJI, but this option is not ethically appropriate. If consensus can be reached on the clinical value of assessing this kind of risk, several preventative measurements can be possible (Table 2). In these patients, all the available weapons should be applied simultaneously as a “preventative package or block” with the hope that synergy among them will decrease the risk much more than any single strategy [58]. In addition, such patients should be cared for by multi-disciplinary teams in “centralized” facilities that specialize in these issues. The Perioperative Orthopedic Surgical Home model offers an example of such an effort [59].

## 3. Perioperative Strategies

A host skin and people working in the operating room environment are a source of bacteria contaminating surgical wounds, tools, and implants (Figure 2). Many studies have demonstrated the significance of operating room quality, including controlling traffic and airflow for the number of surgical site infections (SSIs) [92,93,94,95]. Therefore, measurements contributing to a decrease in the bacterial load during a surgery are extremely worthwhile. Attempts have been made to formulate evidence-based standards for good clinical and logistic practice in orthopedic operating rooms [96,97,98,99]. Educational programs aimed at educating orthopedic surgeons (and all other operation room staff) in perioperative strategies of PJI prevention are ongoing.

### 3.1. Preparation of Operative Field

Skin preparation of the operative field is an essential component of PJI prevention, substantially decreasing the bacterial load of skin origin. Hair removal at the incision site has been analyzed in relation to the risk for SSI/PJI. Current recommendations are to use electric clippers or special depilation creams before TJA surgery. Generally, however, the evidence for routine hair removal is not conclusive [100]. Surprisingly, at least one study showed that hair removal increases the risk for SSI after TKA (total knee arthroplasty) [101]. The probability of PJI was at least three times higher with hair removal compared to surgeries without it (univariate odds ratio [OR] 2.99; 95% confidence interval [CI], 1.24–7.23; multivariate OR 3.09; 95% CI, 1.27–7.50). In France, some surgeons do not recommend hair removal before TJA [100]. Questions also remain unanswered regarding when hair removal should be optimally performed (e.g., the day before the surgery or the day of the surgery).

Preoperatively, antiseptic agents used repeatedly for several days or the night before and/or the morning of surgery have been tested [71,102]. Skin decontamination using a chlorhexidine shower or cloths (other antiseptic agents) could contribute to a decreased PJI risk [103]. The optimal timing or duration of antiseptic agent application is unclear, especially at the shoulder and in case of *Cutibacterium acnes* [104].

The purpose of pre-incisional skin preparation is to reduce radically (within several minutes after application) the number of bacteria at the site of the operating field prior to incision and control for recolonization during surgery [105]. A wide spectrum of paint solutions, including both alcohol- and non–alcohol-based agents, have been certified for this purpose [106], but evidence for a particular product is limited. A recent systematic review and meta-analysis found superior benefit for chlorhexidine compared to povidone-iodine in clean surgery in terms of prevention of postoperative SSIs (risk ratio 0.81; 95% CI, 0.67–0.98) [107]. Concerns persist, however, regarding the effective concentration of chlorhexidine, given a report stating that even with 4% chlorhexidine, skin bacteria could still be cultivated from healthy volunteers [108]. Perhaps along these lines, the results of a recent large randomized controlled trial showed no difference in rates of superficial wound complications between alcoholic chlorhexidine (0.5% chlorhexidine gluconate in 70% ethanol) and alcoholic iodine (1% in 70% ethanol) skin antisepsis [109]. In fact, iodine-alcohol showed greater efficacy in that study in terms of the PJI risk (OR 3.55; 95% CI, 1.20–10.44).

Several methods are available for sealing the surgical field before an incision. One option is to lock skin flora pathogens in place using cyanoacrylate liquid (e.g., Integuseal) before making the incision. However, the evidence in favor of this product in relation to PJI is insufficient to date. In non-TJA studies, SSI rates have not differed significantly between use and non-use of microbial sealants (risk ratio 0.53; 95% CI, 0.24–1.18) [110]. The evidence also is not clearly in favor of iodophor-impregnated adhesive drapes when compared to no adhesive drape in prevention of SSI [111] and PJI [73]. The results of a randomized clinical trial showed a reduction in SSIs when a surgical site preparation solution was reapplied after draping and before the application of iodophor-impregnated incision draping [112].

### 3.2. Operating Room—Technical Parameters, Traffic

The operating room contributes to PJI with the airborne bacterial load. Standards for architectural, technical, and personnel parameters of operating rooms used for TJA surgery have been developed and implemented. These standards describe minimum requirements, including: (i) design (e.g., sizes, places for anesthesiology, operating table, tables for tools, implants); (ii) equipment (e.g., imaging tools, laminar flow, sterilizing facility); (iii) management (“how the operating room area functions as a whole to maintain a sterile environment” including detailed checklists for all regular activities/processes defining personnel/patient/material flows); and (iv) regular assessments/audits of such rooms, guaranteeing together that the environment for surgery is safe for the patient. A large registry-based study found an effect for high-volume, unidirectional, vertical flow ventilation compared with other devices [113]. In contrast, a large study in an Asian population showed no effect on PJI rate after TKA when directly comparing surgeries performed in a laminar airflow operating room to those performed in a room without laminar airflow [114]. In addition, the true cost-effectiveness of ultra-clean air ventilation or laminar airflow systems is not clear [115,116].

### 3.3. Team, Personnel

The key prerequisite for a continual sterile environment is the professional behavior of all operating room personnel during surgery [117]. National [118] or international organizations [119] strictly define the checklists and protocols for personnel/patient/material flows in operating rooms, all of which must be learned, implemented, controlled, and regularly audited. In one study, a positive correlation was found between microbial air contamination and the number of people in the operating room and number of door openings [120]. Global surgery guidelines for the prevention of SSI have been suggested [121], leading to an expectation among surgeons of 100% adherence to pre-established protocols (standards) throughout a surgery, although this bar is a difficult one for all personnel to meet [122,123,124].

The most cost-effective measure in the operating room is hand antisepsis, with a substantial body of evidence showing that modern hand hygiene can reduce the risk for SSIs. Several protocols for hand washing have been developed, tested, and implemented in operating theater practice [125,126,127]. Handwashing for surgery is a complex procedure that is not globally standardized, with a variability among clinical practices at least in terms of the specific washing regime and antiseptic use. Two methods are commonly used for surgical hand washing: i) an alcohol-based hand rub using either a liquid solution, certified gel, or foam-type product; or b) a water-based hand scrub with certified chlorhexidine or povidone-iodine. Manufacturers’ instructions must always be respected. To date, no conclusive evidence highlights the superiority of one method over another for reducing SSI [128].

Surgical team members (including nurses) should have appropriate surgical gowns to decrease skin/hair/body contamination of the operating environment [118]. Some studies have examined the role of body exhaust suits in comparison to traditional surgical gowns, caps, and masks, and the outcomes did not favor modern-type body exhaust suits over standard surgical clothing [129,130]. A sterile surgical helmet system also is not absolutely reliable in terms of preventing contamination [131,132]. In fact, sterile surgical gloves and gowns can be contaminated during longer procedures [133,134,135], and double gloving is recommended in TJA surgery to eliminate contamination of the operating site [136]. Many surgeons change gloves several times per surgery, depending on the operating time or phases of surgery.

### 3.4. Systemic/Local Antibiotics

To date, the use of systemic antibiotics represents one of the most effective approaches to reducing PJI (generally, SSI). Antibiotics are recommended as part of a complex preventative strategy [137]. In no case is the treatment intended as a substitution for other preventative measures.

The U.S. Center for Disease Control and Prevention recommends that prophylactic antibiotics should be administered 1 h prior to surgical incision and repeated in the recommended dose if the surgical time extends beyond 2 or 3 h or in cases of substantial blood loss [7]. The European practice is slightly different [138]. Current guidelines for TJA clinical practice most often recommend cefazolin and cefuroxime for patients undergoing TJA, and vancomycin or clindamycin in those with suspected/proven hypersensitivity to the first-line antibiotics [139]. A universal protocol for antibiotic prophylaxis rather than one tailored for an individual patient should be used in clinical practice [140]. In revision cases, patients should not receive antimicrobial substances for at least 2 weeks before culture sampling to minimize the chance of false-negative culture results [141]. However, at least one small study has shown that preoperative antibiotic prophylaxis does not interfere with the accuracy of tissue culturing [142].

Although most guidelines have recommended the 24-h regime of intravenous antibiotic prophylaxis [139], some alternative protocols have been tested. A *single antibiotic dose* has proved comparable to 24-h and longer regimes in terms of the postoperative rate of PJI [143]. In addition, *oral antibiotics* have been examined in the prevention of PJI, especially in association with an extended period of use and for high-risk patients [78,79,144,145]. Recently, a dual antibiotic strategy has been proposed in response to increased rates of cephalosporin-resistant bacteria. Initial studies have yielded promising outcomes for either a combination of cephazolin with teicoplanin [146] or cephazolin with vancomycin [77]. However, it is too early to widely implement this strategy.

Local antibiotics (antimicrobials) have been proposed and tested for the creation of a more effective local antimicrobial environment in the early postoperative period [147,148,149]. However, there are some concerns related to inducing antibiotic resistance [150,151] and to the effect size in terms of decreased PJI rates after total hip arthroplasty (THA)/TKA [87,152]. At least one study has demonstrated the efficacy and safety of direct intra-wound application of 1 g of vancomycin (plus systemic antibiotics) compared with a control group receiving only systemic antibiotics [153]. There is also support from an experimental study for such a practice [154]. However, some concerns persist about effectiveness [155] and safety because this strategy might increase wound healing complications [156].

Several studies have demonstrated that collagen [157], hydrogel [158,159], or mineral [160] carriers of antibiotics and non-antibiotic agents (e.g., chitosan, antimicrobial peptides) are quite effective in vitro and even in vivo. However, the evidence for using such alternative antimicrobial carriers in preventing PJI remains to be demonstrated in clinical studies.

### 3.5. Intraoperative Care for TJA Patients

Meticulous intravascular catheter disinfection contributes to an overall reduction in SSIs [161]. Several studies have demonstrated the importance of regional anesthesia for the success of TJA at least in terms of complication spectrum/rate [162,163]. A number of factors affect the overall performance of organs and tissues and their ability to clear bacteria during a surgery, even though a clear association between the type of anesthesia and rate of PJI has not been reported. Of relevant factors, the most important is organ/tissue perfusion and oxygenation. Perioperative hypothermia induces poor tissue perfusion, and several strategies have been proposed to avoid it, including preoperative or intraoperative warming [164,165]. A benefit of perioperative hyperoxia in non-critically ill adults has not been demonstrated [166].

### 3.6. Surgeon Performance

The risk for PJI increases with increased operative time, offering more time for bacterial colonization of the operating wound and implant surface [167]. Therefore, an experienced and skilled surgeon is highly necessary to reduce the rate of SSI, including PJI, in combination with other preventative measures. When big data from U.S. clinical practices were analyzed, patients of TKA surgeons who had a shorter operation time had a lower risk for PJI (risk ratio 0.52; 95% CI, 0.43–0.64) than surgeons who typically had a longer procedure duration [168]. Other studies have shown an increased risk for SSI in THA in association with increased operative time [169,170].

An experienced surgeon works quickly and precisely, using tissue-preserving surgery, meticulous clotting of the blood, and other techniques that leave periarticular tissues vital without excessive hematomas. One intervention that every surgeon fully controls is washes prior to wound closure (irrigation). These washes can mechanically and biologically reduce the bacterial load in the wound and available surfaces of an implant [171]. Dilute Betadine wash used at the end of surgery and/or before original implants are placed into the bone bed could reduce the rate of acute postoperative PJI [85,172,173]. At least one study found non-inferiority for chlorhexidine gluconate wash in comparison to dilute Betadine wash [84]. A special suture technique and/or materials also have been suggested to contribute to a reduction in postoperative PJIs [174,175]. On the other hand, PJI rates do not vary when staples are used instead of sutures for skin closure [176]. Overall, it is very difficult to distinguish facts and opinions in the individual surgeon practices. The fear of medicolegal litigation is so considerable that the concept of “less is more” is difficult for many clinical practices to embrace.

### 3.7. Anti-Infective Implant

Many studies have examined a wide range of antibacterial principles and surface finishing/modifications [177,178]. A recent systematic review reported a tendency to a lower PJI rate with silver-coated hip megaprostheses primarily used in tumor indications [179]. These implants have also been used after PJI in patients with extensive bone loss [180]. However, the overall evidence in favor of using silver-coated implants is still insufficient [181]. In addition, one small study reported clinical follow-up (mean 5.6 years) for iodine-coated titanium THAs and TKAs in the treatment of postoperative infections in immunocompromised patients [182].

The newest technologies revolutionize the construction of this kind of implant, combining bacteria killing effectors with physical/chemical sensors for the identification of bacteria/bacterial byproducts on the surface and/or in the vicinity of an implant [183]. When approved and available, these implants could be used in immunocompromised patients with a highly increased risk for PJI, including hematogenous disease. Because the number of these patients is expected to increase, there could be a massive cost savings to the health care system. However, some non-technological obstacles persist. Among these are the unintended coalition of manufacturers, regulatory agencies, and perhaps health care payors. The respective behavior of each of these stakeholders contributes to the overall difficulty, precluding the implementation of smart anti-infective devices in clinical practice. This issue is described in detail elsewhere [184].

## 4. Postoperative Strategies

Postoperative strategies are intended to eliminate risk for PJI associated with bacterial colonization of implant surface/periprosthetic tissues postoperatively (after an implant is encapsulated by vital host tissues). This infective pathway is poorly understood and difficult to prevent in comparison with PJIs associated with intraoperative colonization.

### 4.1. Wound Care

Early wound leakage is a risk factor for PJI [185]. For this reason, care of surgical wounds is an essential part of the early postoperative preventative strategy. Several predictive tools are available for indicating the probability of wound disturbances, including examination of serum albumin, lymphocytes, transferrin, and/or neutrophil/lymphocyte ratio [186]. However, no one of these tests is reliable enough to differentiate risk among patients.

A number of wound dressing products are currently available, with the manufacturers of each one emphasizing its advantages over the others. Some of them have been intended for anti-infective or antibacterial use, employing novel technologies and antibacterial substances [187]. They can be passive, active, and primarily anti-infective (interactive). However, no evidence is available to date favoring any one dressing over another for significantly affecting PJI risk. A compression bandage was found to have no benefit after TKAs [188]; however, negative-pressure wound therapy following THA and TKA has been found to reduce short-term wound healing complications [189]. This approach can be applied especially in at-risk patients [190]. Unfortunately, all of these studies may be biased by study protocols, by the number and type of patients included, and by manufacturers needing to sell their products. The makers also substantially affect practice when they organize education for physicians, physician assistants, and/or nurses in the care for wounds.

### 4.2. Measures Against Hematogenous and Directly Spreading PJI

After a wound is healed, hematogenous contamination and direct spreading (“per continuitatem”) of an artificial joint remain the only pathways for new access by pathogenic bacteria [191,192]. To prevent these PJIs, we should theoretically target all potentially dangerous sources of bacteria, but no one knows which these are. Another assumption is that the capsule around an artificial joint must lose its protective function.

Traditionally, the sources of clinically important bloodstream contamination are dental, urinary, gynecological, renal, or bowel surgical infections/procedures. Recent analyses argue against routine prophylaxis before dental procedures [193,194]. No or insufficient evidence is available in relation to other interventions potentially leading to substantial bacteremia (abdominal, gynecological, urological procedures). The current recommendations on antibiotic prophylaxis after TJA are listed in Table 3. Scientifically, the task is relatively simple: to prove a substantial decrease in PJI after administration of antibiotics in these cases compared to those who did not receive this prophylaxis. Such studies must involve modern informatic technologies, connecting data from registries of TJA with other digital sources of information for hundreds of thousands of patients.

## 5. A Case for Quality Improvement into Practice

Through many years of practice, a number of strategies have been tested, all aimed at reducing the risk for PJI development, applied pre-, peri-, or postoperatively. Even though their educational impact has been enormous based on the number of their secondary products (e.g., papers, lectures, guidelines), the overall risk for PJI ranges from 0.5% to higher depending on the study location, design, and/or a particular group of patients who were treated by TJA. The risk-standardized PJI event ratio can be calculated to adjust for different volumes and other factors influencing the inappropriate comparison between locations [204]. Taken together, we have a relatively solid body of knowledge about “anti-infective” interventions; however, we perhaps fail in implementing these strategies in clinical practice. This situation is nothing new in clinical medicine.

Quality improvement (QI) is defined as a systematic continuous approach that ultimately provides better outcomes for patients via a set of methods, their appropriate communication, re-assessment of their effect, and their correction if required [205]. Health care professionals frequently feel resistance to QI workers and limit collaboration with them. The reasons for this resistance are many and mixed and beyond the scope of this review. The most important question is whether the risk for PJI could be decreased to zero in all departments through a tighter collaboration with QI departments. In other words, can a multi-factorial event like PJI be eliminated using QI approaches? This question should be tested because improvement in healthcare is estimated to be 20% technical and 80% human [205]. If this estimation is true, then there is a large window for improvement.

Healthcare organizations use a clinical audit to track and confirm the organization’s adherence to current written standards. The primary intent for service evaluation is to assess the performance of current patient care [205]. However, both of these regular activities may co-exist with an increased rate of PJI. A constraint on this possibility is that the health care service/insurance companies often have a strict predefined threshold for penalizing hospitals with PJI numbers that exceed this value. In fact, one of the critical functions of national registries is to identify the “problematic departments” (not only implants). Red flags about a service performance enable local authorities to initiate the QI process to change clinical practice. Details related to a particular QI approach are out of the scope of this review, and we recommend some of the recent relevant literature [206,207,208].

## 6. Who Is the Best Provider?

Theoretically, the best provider can implement a complex body of preventative measurements, verify their effects continuously, and upgrade the system according to new evidence as needed (Figure 3). In practice, the winners are the providers reporting the lowest PJI rate after adjustment for the number of at-risk patients. Based on these data, patients should not be admitted to hospitals that cannot adopt the highest standards of prevention that guarantee the lowest risk for PJI. Let us assume that accurate data are available for a department or a surgeon. The reality is that every provider has some rate of PJI regardless of the quality of preventative measures applied. We can speculate about the reasons underlying each PJI. Does PJI occur only in cases when the otherwise functioning systems fail accidentally? Or is it that the system prevents infection in low- or middle-risk patients and cannot cover only those who are susceptible to infection?

Several researchers have sought to summarize the evidence from studies with large heterogeneity in terms of patients, methodology, confounding factors, or outcomes [209,210,211]. Adding a new intervention to the relatively robust multi-component and multi-level preventative system gives information only on the performance of the whole system after adding a new intervention, not on the pure performance of an individual measurement. Therefore, it is difficult to evaluate the individual contribution to the performance of a system. In addition, the performance of the whole system may be so effective that it leads to an overestimation of the individual intervention or even adaptation to mistakes. As a result, some workplaces can maintain a very low rate of PJI, whereas others show a higher rate of PJI using the same approach to prevention.

The secret to success probably consists of a well-balanced combination of preventative interventions that will result in a reduction in the number of PJIs in the population and the probability of PJI in a patient [212]. Whether finding it will require trial and error or be facilitated by more modern tools remains to be seen. Still, it is possible to identify the fault in advance when processing sufficiently large samples of patients. If joint replacement registries uncover a higher number of PJIs in some workplaces, it can be possible to infer that there is a problem in operation rooms or in early postoperative care based on a particular patient portfolio.

## 7. Contribution of Machine Learning to Prevention of PJI

Machine learning (ML) references a group of methods for training software algorithms to learn from and act on new data to continuously improve health care performance [213]. Its utility for further progress in surgery is clear and unquestioned [214]. Here, we would like to emphasize the potential of ML to reduce the number of PJIs by mining new knowledge from the huge amount of data collected for individual patients (Figure 4) in every country. To date, no single study has demonstrated the contribution of a particular ML method for preventing PJI. This gap is at least in part a result of the strict regulatory approach to artificial intelligence technologies. When intended to prevent a disease (complication), ML-based software is defined as a medical device under the Food, Drug, and Cosmetic Act with regulatory steps which must be fulfilled before implementation into a clinical practice. Most ML-based products could be developed via the 510(k) pathway or through the de novo pathway [213]. However, concerns have arisen, for example, about a change in product outputs after the product is distributed. The primary reason for these concerns is the potential for “after learning” self-corrections that could lead the device to provide useless or even harmful recommendations compared to regulatory approval. Therefore, a post-approval regulatory strategy is required to guarantee the continual safety of these products. At the time of writing, it is important to note that these technologies are a reality and that orthopedic surgeons in collaborations with bioinformaticians and engineers can apply them to all PJI topics, including prevention. Big data and informatics are central to such efforts and can deliver feedback to surgeons and their teams so that they can learn from their experiences and refine their practices. Doing so will require international effort because enormous numbers of TJA patients would have to be included in such a project.

## 8. A Step towards Precision Prevention of PJI?

A very low microbial load is a critical requirement for each TJA surgery because without microbes (bacteria or fungi), PJI cannot develop. To achieve this aim, multiple measurements should be applied simultaneously in all patients (routine standards for everyone). This approach, however, has generated the relatively wide range of PJIs reported in the literature, and there is a suspicion that the numbers of PJI are underestimated. Certainly, not every surgery is performed under the same conditions. Variability in the quality standards or adherence to these standards underlies some PJIs, but these factors make up only part of the truth. Another part of the reality is that PJI is perhaps unpreventable in some patients regardless of the environment in which the surgery is performed. For this reason, current standards for TJI may not be able to protect all patients who are at risk for PJI.

Precision prevention of PJI describes delivery of the *precise preventative content to a particular patient at an optimal time* [215]. In other words, it is about tailoring prevention to a biological as well as non-biological risk profile of the individual patient. However, the understanding of how patients differ in terms of their susceptibility to PJI before TJA remains insufficient despite the number of studies published in the last 10 years. These studies have identified several risk factors that can be largely categorized as modifiable and non-modifiable [211,216,217,218]. Of course, it is tempting to change all avoidable factors before a surgery to diminish the overall PJI risk. However, an optimal model for precise stratification of patients is not available in clinical practice.

Theoretically, the preventative interventions should be matched to a particular patient according to the individual preoperatively calculated risk for PJI [219,220]. For at-risk patients, depending on their estimated risk for PJI, an extra-preventative package could include, e.g., surgery in a special “anti-infective” room, prolonged antibiotic prevention, placement of a local antibacterial carrier around the implant, and implantation of an anti-infective implant. However, no study has demonstrated the success of precision prevention in the field of TJA. Of note, any decrease in PJI demonstrated in well-conducted studies should be non-trivial to justify the overall increased costs associated with such an intervention.

## 9. Conclusions

The prevention of PJI is an absolutely essential part of TJA clinical practice because all patients have the right not to be harmed during TJA. When complex preventative measures are fully applied and adhered to in clinical practice, PJI should be preventable in almost all cases. Thus, the secret of clinical success (i.e., negligibly low infection rate) lies in an optimal combination of critical preoperative and perioperative measurements. Strategies relying on the creation of a bacteria-free environment around an implant during the whole perioperative period are a cornerstone of a successful TJA clinical practice. Sufficient evidence supports systemic antibiotic prophylaxis and some other traditional preventative measures. Here, we postulate that all future strategies will focus on continuous data and quality management. The optimal preventative configuration has to be continuously designed/redesigned according to the actual rate of PJI and its appropriate analysis. In this sense, prevention will always be a systematic work in progress. The outcomes of such PJI-oriented QI projects have to be monitored and analyzed. If a particular preventative configuration shows excellent efficacy, then it should be implemented, trained on, and controlled widely across hospitals globally. Patients, payors, and regulatory authorities should insist on adherence to these up-to-date preventative standards.

## Figures and Tables

**Figure 1 jcm-09-02190-f001:**
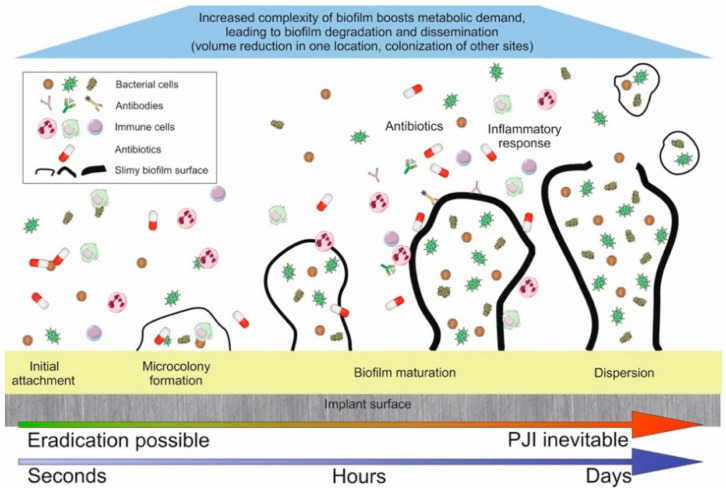
Formation of biofilm. PJI, prosthetic joint infection.

**Figure 2 jcm-09-02190-f002:**
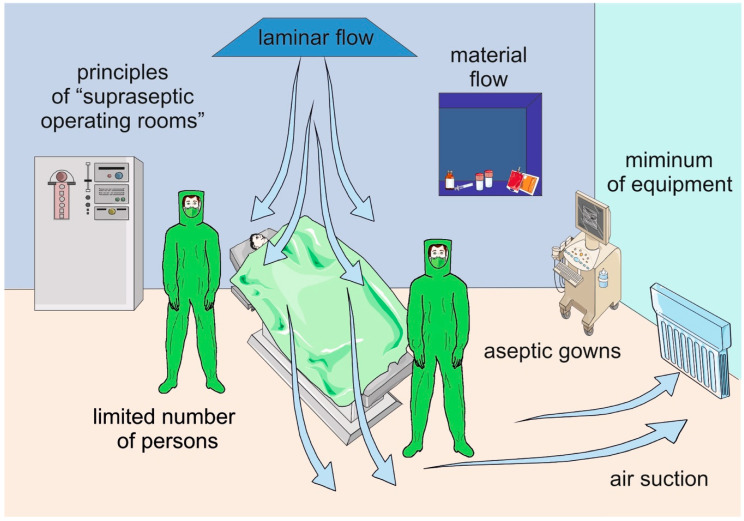
Preventative measures associated with the operating room and aimed at reducing the magnitude of the personnel/patient-derived bacterial load.

**Figure 3 jcm-09-02190-f003:**
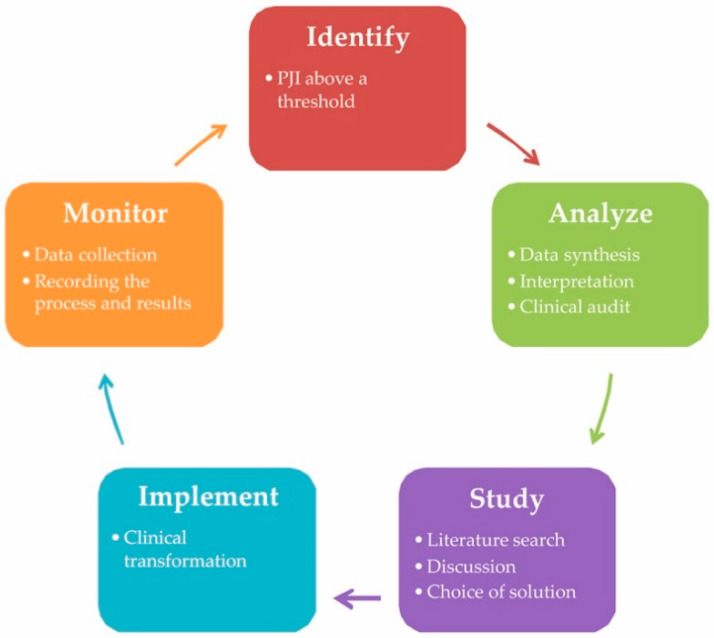
Risk management cycle for continual monitoring of quality of practice. The goal of clinical improvement is to control for quality indicators (here, the rate of PJI). An essential component is a systematic approach to the clinical reality based on continual data collection and analysis. The results of this approach should lead to an understanding of how to address the pertinent indicator through effective correction of clinical processes, followed by monitoring. PJI, prosthetic joint infection.

**Figure 4 jcm-09-02190-f004:**
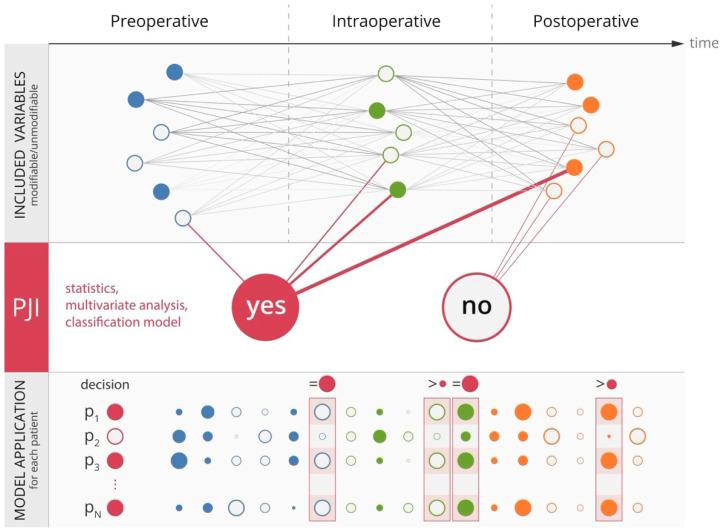
Formalization of continuous data collection and processing. The upper layer, “included variables,” represents long-term follow-up/measurement of parameters that can influence each other. The weight of the parameters is indicated by the intensity of the gray lines and the color in the circles. Weight can also equal 0, which would imply that the attributes do not influence each other. The middle layer, “PJI,” corresponds to the formation of the classification model, initially according to the most recent definition of PJI. The output of this layer is a “trained” classification model used for classification of all patients, including future ones. The most significant attributes and their limits, which take the greatest part in decision-making (represented in dark red), are set in particular classification classes (PJI yes; PJI no). The lower layer, “model application,” represents the practical use of the “trained” model for classification of patients. Selected key classification parameters (PJI yes) are shown here, illustrating which values these attributes should acquire to result in the overall classification of “PJI yes.” Of note, the measured values and deviations in classification can lead to recalculation of the model, i.e., returning to the upper layer. PJI, prosthetic joint infection.

**Table 1 jcm-09-02190-t001:** Comorbidities and medications that are risk factors for PJI.

Variable	Level of Evidence	Estimated Size Effect	Population Effect
Systemic malignancy	Individual study [38] SR [39]	OR 3.1; 95% CI, 1.3–7.2 RR 1.52; 95% CI, 0.98–2.34	Limited
Chronic kidney insufficiency	Individual studies [40,41,42,43]	5-year PJI rates, 8.5%; in transplant patients, 4.5% [44]	Low #
Severe hepatic disease	SR [45]	Revision rate, 4% (vs. 0.2% in controls); PJI in 70%; mean infection rate, 4.1% (for urgent cases, 8.6%) ^¶^	Limited
Rheumatoid arthritis	SR, MA [46]	OR 1.89, 95% CI, 1.34–2.66 ^¶¶^	Moderate
Risk of biologics	Individual study [47]	Compared to a 2.14% 1-year cumulative incidence of PJI with abatacept, predicted incidence ranged from 0.35% (95% CI, 0.11–1.12) with rituximab to 3.67% (95% CI, 1.69–7.88) with tocilizumab	Low to moderate
BMI (≥30) BMI (≥40/<30)	SR, MA [48]	RR 2.22; 95% CI, 1.67–2.96 RR 8.48; 95% CI, 3.47–20.71	Strong
Malnutrition	SR + MA [49]	OR 3.58, 95% CI, 1.82–7.03	Moderate
Diabetes	SR/MA for SSI/for PJI [39]	RR 1.74; 95% CI, 1.45–2.09	Strong
Smoking	SR/MA [50]	OR 2.02; 95% CI, 1.47–2.77	Strong *
Lymphedema	Individual study [51]	Revision HR 6.19 (95% CI, 2.22–17.23)	Limited to low #
HIV + patients	SR [52]	Risk ratio 3.31; 95% CI, 1.18–9.29	Limited
Hypothyroidism	Individual study [53] Individual study [54]	OR 2.04; 95% CI, 1.02–4.08 ^TEA^ OR 1.32; 95% CI, 1.03–1.69 ^TAA^	Low
Prior joint surgery	SR [39]	RR 2.98; 95% CI, 1.49–5.93	Strong
Prior PJI in another joint	Individual study [55]	HR 3.3, 95% CI, 1.18–8.97 (patients on chronic suppression: HR 15)	NR
Previous steroid administration	SR [39]	RR 1.68; 95% CI, 1.26–2.25	Low to moderate
Peripheral vascular disease	Individual study [54]	OR 2.46; 95% CI, 1.87–3.22	Low to moderate #

Population effect: limited, occurring in <1% of all patients indicated for TJA; low, occurring in 1–5%; moderate: in 5–10%; strong: >10%; BMI: body mass index; CI: confidence interval; HR: hazard ratio; MA: meta-analysis; NR: not reported; OR: odds ratio; PJI: prosthetic joint infection; RCT: randomized controlled trial; RR: relative risk; SR: systematic review; SSI: surgical site infections; TJA: total joint arthroplasty; *: in dependence on the society/region/state; #: in dependence on severity/stage of a disease; ^¶^: only for total hip arthroplasty; ^¶¶^: only for total knee arthroplasty; ^TEA^: only for total elbow surgery; ^TAA^: only total ankle arthroplasty.

**Table 2 jcm-09-02190-t002:** Examples of preventative measurements that could contribute to decreased risk for PJI in the general population.

Intervention	Description of Effect
Pre-/perioperative serum glucose control, hemoglobin A1c, fructosamine	Optimizing for patients with unstable diabetes could target (i) wound healing; (ii) SSI; and (iii) re-admission rate [60,61,62,63].
Reducing BMI	Optimizing weight for patients with overweight could target (i) wound healing; (ii) SSI; and (iii) PJI rate [64,65,66]. Patients with BMI >35 are advised to lose weight prior to surgery; however, candidacy restriction is not appropriate because these patients have no other options for pain relief.
Treatment of all preoperative infections	To reduce the chance of the development of hematogenous and/or directly spread PJIs.
Staphylococcal decolonization (nasal, skin)	Elimination of *S. aureus* (MRSA) carriers from TJA surgery could contribute to reductions in PJI [57]; however, some patients may be carriers even after decontamination [67,68].
Discontinuation of immunosuppressive therapy	To reduce the effect of therapy on the capacity and efficacy of the immune system; various recommendations: (i) methotrexate off (1 wk preop.; 2 wks postop.); (ii) anti-TNF agents off (2–8 wks preop.; 2–4 wks postop.) [7]; described in detail elsewhere [69].
Preoperative wash/cloth with antibacterial substances	To reduce bacterial skin load; all chlorhexidine, dilute povidone-iodine solutions can contribute to reduced risk for SSI/PJI [70,71,72].
Antibacterial incisional drape	To eliminate the residual bacterial skin load after routine skin preparation [73,74,75].
Novel strategies for systemic/local antibiotics/antimicrobials	Extension and/or prolongation of the antibacterial effect of ATBs via dual [76,77] or extended ATBs [78,79]; limited evidence for intrawound ATBs [80].
The best operating rooms/surgeons *	To decrease the intraoperative bacterial load via an experienced surgeon, team, and aseptic theater [81,82,83].
Intraoperative wound wash of antimicrobials	To decrease the intraoperative bacterial load [84,85].
Implants with antibacterial surfaces	To improve resistance of an implant against bacterial adhesion via antibacterial hydrogel [86], other antibacterial carriers [87,88] or silver coating [89].
Anti-staphylococcal vaccine	To increase efficacy of the immune system to eradicate *Staphylococcus* spp. intra-/postoperatively [90,91].

* high-volume surgeons working in the best operating rooms; ATB: antibiotic; BMI: body mass index; MA: meta-analysis; MRSA: methicillin-resistant *Staphylococcus aureus*; PJI: prosthetic joint infection; postop.: postoperatively; preop.: preoperatively; SSI: surgical site infection; SR: systematic review; TJA: total joint infection; TNF: tumor necrosis factor.

**Table 3 jcm-09-02190-t003:** List of events that could arise after TJA surgery and recommendations potentially linked to reducing the risk for PJI.

Event	Recommendation	Ref.
Dental intervention	Evidence against routine ATB prophylaxis after TJA, only in special clinical situations	[193,195,196]
Abdominal surgery	Inconclusive evidence, ATBs are indicated, apart from situations always requiring ATB therapy, also for advanced forms of acute appendicitis, perirectal abscess, invasive endoscopy procedures on the colon, soft tissue phlegmon or abscess, surgical treatment of venous ulceration and pressure sores, and limb amputation	[197,198,199]
Cardiological interventions	Patients with TJA should not receive ATBs before cardiovascular interventions	[199]
Dialysis	No recommendation in relation to a combined risk from the chronic vascular approach and end-stage renal insufficiency	[40]
Urinary tract intervention, infection	Inconclusive evidence, however, a risk-associated procedure on the urogenital system (endoscopic or open surgery, prostate gland biopsy, extracorporeal lithotripsy) could be subject to ATB prophylaxis	[197,198,199]
Skin infections	Treat emergently all erysipelas as it can affect previously healthy TKA	[199,200,201,202]
Postoperative immunosuppressive therapy	Reduce doses of glucocorticoids because prednisone increases risk for a postoperative infection (OR 1.59, *P* < 0.001)	[203]
	Propensity-adjusted HR 1.36 (95% CI, 0.90–2.04) for 5–10 mg and 1.86 (95% CI, 1.02–3.37) for >10 mg	[47]
	described in detail elsewhere	[69]

ATB: antibiotic; PJA: prosthetic joint infection; THA: total hip arthroplasty; TKA: total knee arthroplasty, HR: hazard ratio.

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
