# Peer review of "Prevention of Prosthetic Joint Infection: From Traditional Approaches towards Quality Improvement and Data Mining"

_jcm, 2020, doi:10.3390/jcm9072190_

Round 1
Reviewer 1 Report
The manuscript is an excellent review of preventive measures for avoiding PJI. In my opinion, only minor changes are necessary:
1-Line 77: Staphylococci does not need to be written in italics, because it is not an scientific name.
2-Line 118: I think that the title (aimed to comorbidities that decrease immune system efficacy) needs a change, because some conditions (peripheral vascular disease, prior joint surgery) do not affect immune system. I suggest to change to comorbidities that are a risk factor for PJI.
Line 138: In the table it will be of interest to add a column about the level of evidence.
Line 154: In the figure could be of interes adding laminar flow.
Line 233-235: Include a paragraph about the use of alcogoloc solucions and gels in surgical handwashing.
Line 246: In the subheading about antibiotics, include a paragraph about the importance of the use of antibiotics before surgery in culture results, and about the implications of chemoprophylaxis in culture results.
Line 340: In the subheading about anti-infective implants include only experience with available implants. There is an enormous number of articles about research in the materials field regarding antimicrobial properties, so I think that it is better to speak about the available ones.
Line 418: "the winners are the providers reporting the lowest PJI rate." It is true, but it depends also on the complexity of the patients. As higher is the complexity, as higher is the risk of PJI, so many high-complexity big centers have higher rates than low-complexity small hospitals, and it does not mean that the big ones are worse than the small ones. Please try to include this concept in the text in order to avoid misinterpretation.
Author Response
AUTHORS RESPONSES TO THE COMMENTS OF THE REVIEWERS
At first, we thank all the reviewers for thoroughly reading and critiquing our manuscript, and appreciate all the comments and suggestions.
Reviewer 1.
The manuscript is an excellent review of preventive measures for avoiding PJI. In my opinion, only minor changes are necessary:
1-Line 77: Staphylococci does not need to be written in italics, because it is not a scientific name.
Response: Thank you, the change has been made (line 80 in the new MS version).
2-Line 118: I think that the title (aimed to comorbidities that decrease immune system efficacy) needs a change, because some conditions (peripheral vascular disease, prior joint surgery) do not affect immune system. I suggest to change to comorbidities that are a risk factor for PJI.
Response: Thank you, we agree with your suggestion – the title has been changed (line 120 in the new MS version):
Table 1. Comorbidities and medications that are risk factors for PJI.
Line 138: In the table it will be of interest to add a column about the level of evidence.
Response: Thank you for your suggestion. At the beginning of the writing we also felt that the level of evidence should have been included in the table, at least using a simple statement like: “a systematic review of RCTs or RCTs are in favor of using a particular intervention – yes/no” in the right column. However, this does not bear any clinically useful information on the size of the effect; consistency of findings can also be questioned as studies against particular statements are available too. This review is only a narrative one, and Table 2 is merely of illustrative nature. In order to clarify the information in the table, we have re-arranged it (page 5 in the new MS version).
Line 154: In the figure could be of interest adding laminar flow.
Response: Thank you, the figure has been changed (Fig.2).
Line 233-235: Include a paragraph about the use of alcoholic solutions and gels in surgical handwashing.
Response: Thank you for the suggestion, a text has been added (lines 229–235 in the new MS version) as follows:
Surgeon handwashing is a complex procedure, not uniform in the world, with a variety in clinical practice at least in terms of a particular washing regime, or antiseptic means. Manufacturers’ instructions must always be respected. Basically, two methods are commonly used for surgical hand wash: i) alcohol-based hand rub with either alcohol solutions, or a certified gel, or foam-type product; b) water-based hand scrub with certified chlorhexidine or povidum-iodine. To date, there is no conclusive evidence on the superiority of one method over another for reducing SSI [128].
Line 246: In the subheading about antibiotics, include a paragraph about the importance of the use of antibiotics before surgery in culture results, and about the implications of chemoprophylaxis in culture results.
Response: Thank you, a text dealing with this significant issue has been added (lines 256–259 in the new MS version) as follows:
In revision cases, patients should not receive antimicrobial substances for at least two weeks before culture sampling in order to minimize the chance for false-negative results of this examination [137]. In fact, at least one small study shows that pre-operative antibiotic prophylaxis does not interfere with the accuracy of tissue culturing [138].
Line 340: In the subheading about anti-infective implants include only experience with available implants. There is an enormous number of articles about research in the materials field regarding antimicrobial properties, so I think that it is better to speak about the available ones.
Response: Thank you, we have changed the text (lines 313–330 in the new MS version) as follows:
3.7. Anti-infective Implant
Many studies have examined a wide range of antibacterial principles and surface finishing/modifications [173,174]. A recent systematic review reported a tendency to a lower PJI rate with silver-coated hip megaprostheses primarily used in tumor indications [175]. These implants also have been used after PJI in patients with extensive bone loss [176]. However, the overall evidence in favor of using silver-coated implants is still insufficient [177]. In addition, one small study reported clinical follow-up (mean 5.6 years) for iodine-coated titanium THAs and TKAs in the treatment of postoperative infections in immunocompromised patients [178].
The newest technologies revolutionize the construction of this kind of implant, combining bacteria killing effectors with physical/chemical sensors for identification of bacteria/bacterial byproducts on the surface and/or in the vicinity of an implant [179]. When approved and available, these implants could be used in immunocompromised patients with a highly increased risk for PJI, including hematogenous diseases. Because the number of these patients is expected to increase, there could be a massive cost savings to the health care system. However, some non-technological obstacles persist. Among these is the unintended coalition of manufacturers, regulatory agencies, and perhaps health care payors. The respective behavior of each of these stakeholders contributes to the overall difficulty, precluding implementation of smart anti-infective devices in clinical practice. This issue is described in detail elsewhere [180].
Line 418: "the winners are the providers reporting the lowest PJI rate." It is true, but it depends also on the complexity of the patients. As higher is the complexity, as higher is the risk of PJI, so many high-complexity big centers have higher rates than low-complexity small hospitals, and it does not mean that the big ones are worse than the small ones. Please try to include this concept in the text in order to avoid misinterpretation.
Response: Thank you for your comment. It is extremely important to avoid misinterpretation/ misunderstanding of this particular part. We have re-written this part (lines 405–406 in the new MS version) as follows:
In practice, the winners are the providers reporting the lowest PJI rate after adjustment on the number of at-risk patients.
Reviewer 2 Report
Bacterial infection at the site of joint prosthesis remain a major complication of prosthetic surgery. In that, the review by Gallo and Nieslanikova addresses an important issue. They summarize the current knowledge of biofilm formation of implants, the means taken to prevent bacterial infections, and they point out the limitations of those measures. The review is extensively underscored by literature, which makes is a valuable tool for surgeons and health personnel as well. It, however, does not provide a critical evaluation of the data – only a summary. Moreover, new insights or novel procedural methods are not included.
Specific issues: Table 2 – which addresses possible means to reduce infection – suggests “immunization”. The explanation, however, is not correct: immunization (= vaccination) generates a specific immune response e.g. by antibody formation; “increasing the efficacy of the immune response” or “modifying” is – contrary to common belief - not necessary, and is contraindicated considering the fact that the major tissue damage is caused by the inflammatory process that is elicited by the local immune response. Actual “immunodeficiency” is a very rare condition.
Author Response
AUTHORS RESPONSES TO THE COMMENTS OF THE REVIEWERS
At first, we thank all the reviewers for thoroughly reading and critiquing our manuscript, and appreciate all the comments and suggestions.
Bacterial infection at the site of joint prosthesis remain a major complication of prosthetic surgery. In that, the review by Gallo and Nieslanikova addresses an important issue. They summarize the current knowledge of biofilm formation of implants, the means taken to prevent bacterial infections, and they point out the limitations of those measures. The review is extensively underscored by literature, which makes is a valuable tool for surgeons and health personnel as well. It, however, does not provide a critical evaluation of the data – only a summary. Moreover, new insights or novel procedural methods are not included.
Response: Thank you for your valuable comment. At first, we have changed the title of the review so as to be clearer (lines 1–4 in the new MS version).
Prevention of Prosthetic Joint Infection: from Traditional Approaches towards Quality Improvement and Data Mining
Specific issues: Table 2 – which addresses possible means to reduce infection – suggests “immunization”. The explanation, however, is not correct: immunization (= vaccination) generates a specific immune response e.g. by antibody formation; “increasing the efficacy of the immune response” or “modifying” is – contrary to common belief - not necessary, and is contraindicated considering the fact that the major tissue damage is caused by the inflammatory process that is elicited by the local immune response. Actual “immunodeficiency” is a very rare condition.
In addition, we have re-written a particular part of Table 2 (page 5 in the new MS version).
|
Anti-staphylococcal vaccine |
To increase efficacy of the immune system to eradicate particular skin/airborne bacteria intra-/postoperatively [90,91] |
Reviewer 3 Report
Gallo et al. made an intensive review on infection prevention strategies in hip and knee PJIs.
Although the authors give a nice overview on a lot of different aspect on infection prevention, and put a lot of effort into this, in my view the study does not add much to current literature.
Infection prevention measures have been extensively prescribed in WHO guidelines, and more recently in the international consensus meeting documents (ICM 2018, Philadelphia). In addition, the document is hard to read. Especially in the beginning, the outline is too extensive, with many side paths addressed, and with a lack of clear focus. This demotivates the reader to continue reading.
I would probably only advise publication if the review would be more comprehensive, with the focus on "new strategies" like the authors claim in the title. The majority of the topics the authors described now are not new at all.
Author Response
AUTHORS RESPONSES TO THE COMMENTS OF THE REVIEWERS
At first, we thank all the reviewers for thoroughly reading and critiquing our manuscript, and appreciate all the comments and suggestions.
Gallo et al. made an intensive review on infection prevention strategies in hip and knee PJIs.
Although the authors give a nice overview on a lot of different aspect on infection prevention, and put a lot of effort into this, in my view the study does not add much to current literature.
Infection prevention measures have been extensively prescribed in WHO guidelines, and more recently in the international consensus meeting documents (ICM 2018, Philadelphia). In addition, the document is hard to read. Especially in the beginning, the outline is too extensive, with many side paths addressed, and with a lack of clear focus. This demotivates the reader to continue reading.
I would probably only advise publication if the review would be more comprehensive, with the focus on "new strategies" like the authors claim in the title. The majority of the topics the authors described now are not new at all.
Response: Thank you for reading of our manuscript. We feel very sorry for difficulties you had reading our manuscript. It is the greatest effort of the authors to achieve good readability of the text. On the other side, it is not easy to write a review on the issue of PJI prevention in a completely new format. We wanted to show what was known (“the current state”) first, and then present our observation that despite an enormous body of research performed in the last decade, it is very difficult to achieve a substantial progress in the field of prevention of PJI without using new technological platforms. We can closely monitor our clinical practices using quality improvement measurements. We can systematically collect data on every patient undergoing TJA, every surgery. Sharing data about all infected/non-infected individuals within the public and private sectors brings epidemiological benefits. However, the complexity of the clinical situation (patient, preoperative preparation, operating room, surgeon etc.) determines the complexity of any computational model required for optimization of a preventative strategy. Even though the TJA-related data are not structured homogenously like for instance imaging data, there are now methods enabling analysis of highly heterogeneous and sometimes inconsistent patient/surgery-related data. Uncertainty related to the development of PJI may be reduced only using a multi-pack of preventative interventions and employing big data including EHRs. And this is the novelty available now, indicated also in the title of our manuscript.
We have read the manuscript thoroughly once again and made a number of changes including shortening of some parts. However, we feel that our manuscript will not be well understandable without showing limitations of the traditional preventative methods in the first parts of the manuscript. Therefore, the basic structure of the manuscript remained as it was in the original draft.
Round 2
Reviewer 2 Report
Many thanks for the revised manuscript.The first aspect has been adequately corrected. However, the second comment has not been addressed.
(As stated above, the authors summarize the current knowledge on prosthetic joint infection and the efforts to reduce the risk of infection. The extensive literature makes the review a valuable tool for orthopedic surgeons and health personnel. However,given the very specific topic, I am not convinced that the review is of interest for the majority of readers, particularly because there are not real “new insights” and no conclusive recommendations for better “good practice.“)
Author Response
Thank you for your comments. Based on your previous comment we changed the title of the review to clarify our original intention. Therefore, it is now clear that our review is about data and quality management of traditional preventative measures. You are right that the readers might get a misleading impression from the original title.
In addition, we made several changes in the conclusion of the review to more clarify the key points.
Reviewer 3 Report
ok for me
Author Response
We read the manuscript thoroughly again and made several grammatical and typographical corrections. Thank you.